# A Review of the Emerging Risks of Acute Ammonia Nitrogen Toxicity to Aquatic Decapod Crustaceans

Wang Lin [1,2], Huimin Luo [1], Jingyi Wu [1], Tien-Chieh Hung [3], Beibei Cao [1], Xiangli Liu [1], Jifeng Yang [4] and Pinhong Yang [1,*]

1   College of Life and Environmental Sciences, Hunan University of Arts and Science, Changde 415000, China
2   Institute for Ecological Research and Pollution Control of Plateau Lakes, School of Ecology and Environmental Science, Yunnan University, Kunming 650500, China
3   Department of Biological and Agricultural Engineering, University of California, Davis, CA 95616, USA
4   Chemistry and Material Engineering College, Hunan University of Arts and Science, Changde 415000, China
*   Correspondence: yph588@163.com

**Abstract:** Waterborne ammonia is becoming one of the most notorious pollutants in aquatic habitats and has been shown to induce a range of ecotoxicological effects on aquatic animals. High ammonia concentrations occur mainly in intensive aquaculture systems, and effective wastewater treatment and agricultural systems are necessary to treat excessive nitrogenous compounds. Ammonia can enter aquatic decapod crustaceans through their gills, thereby reducing the oxygen-carrying capacity of blood cells and damaging the structures of organs such as the gills and hepatopancreas. This ultimately results in oxidative stress, immunotoxicity, and high mortality. Crustaceans have the ability to exert detoxification functions against ammonia stress by regulating the permeation of ammonia and related nitrogenous compounds through membranes. To the best of our knowledge, a comprehensive review of the acute toxicity of ammonia to crustaceans is lacking. The present review focuses on the literature on the problems and mechanisms concerning ammonia-induced acute toxicity and aims to synthesize the knowledge of the relationship between ammonia stress and defense responses in crustaceans (mainly shrimp and crabs). This review also emphasizes the uptake, elimination, and detoxification of ammonia in crustaceans.

**Keywords:** ammonia nitrogen; $LC_{50}$; aquatic crustaceans; oxidative stress; detoxification mechanism

## 1. Introduction

The consumption of aquatic animals such as fish and crustaceans is increasing as the standard of living improves [1]. Aquatic organisms are a good source of protein and contain multiple amino acids, essential minerals, and vitamins [2]. Aquatic decapod crustaceans (e.g., shrimp and crabs) are not only important as fishing objects but also crucial aquaculture species in coastal and inland areas, and they have made important contributions to the social and economic development of those areas [3]. The Chinese aquatic organism market is becoming one of the largest markets in the world [4]. The production of crustaceans is growing rapidly with the increase in market demand. According to official statistical data, the aquaculture and capture fisheries production of aquatic crustaceans in China were 6.44 million tons and 1.98 million tons in 2021, respectively, which exhibited increases of 6.73% and 0.60% compared to the year before [5].

For intensive aquaculture systems, uneaten high-protein feed, aquatic animal carcasses, and other debris accumulate rapidly, but degrade slowly, at the bottom of the pond, resulting in a large number of inorganic nitrogen compounds. These compounds pose a risk to the health of aquatic crustaceans [6,7]. In recirculating aquaculture systems, the situation is even more serious since crustaceans can easily encounter high levels of nitrogenous waste [8]. Ammonia is one of the most common nitrogenous pollutants in the aquaculture environment and is mainly produced by, and poisonous to, aquatic animals. Ammonia

normally exists in water in two forms: un-ionized ammonia ($NH_3$) and ammonium ions ($NH_4^+$) [9,10]. Un-ionized ammonia is more toxic than ammonium ions to fish, which is attributed to the lipid solubility and lack of charge in the former form [11,12]. $NH_3$ can diffuse across gill membranes and result in a decreased oxygen-carrying capacity of the blood cells, damaging the structure and function of organs such as the hepatopancreas in crustaceans [13,14]. Increased pH and water temperature can aggravate ammonia's aquatic toxicity because of the increased ionization fraction of $NH_3$ in water [15].

The toxic effects of ammonia on aquatic crustaceans mainly include three aspects. First, exposure to external ammonia increases ammonia levels in the blood and body fluids, and $NH_4^+$ competes with $Na^+$ for passive cation diffusion in the gills, thus affecting osmoregulation [16]. Second, ammonia stress could cause lamellar fusion and epithelial ascent in the gills, leading to the complete collapse of gill lamellae and the interruption of gas exchange [17,18]. Lastly, ammonia exposure results in oxidative stress and endoplasmic reticulum (ER) stress, subsequently causing cell membrane damage, cell apoptosis, and even cell death [14]. Apart from the above-mentioned toxic effects, excessive ammonia stress could eventually induce the high mortality of aquatic crustaceans. The semi-lethal concentration ($LC_{50}$) is the most intuitive indicator of the acute toxic effect of ammonia nitrogen on crustaceans.

Several studies have revealed that high ammonia concentrations are toxic to aquatic organisms, which can result in oxidative stress, immune suppression, pathogen susceptibility, and even high mortality [19–21]. A comprehensive review of the acute toxicity as well as the physiological consequences of ammonia in aquatic crustaceans is lacking. In this review, the research progress on ammonia-induced acute toxicity to aquatic crustaceans is reviewed so as to provide guidance for further toxicological research on ammonia. The uptake, intoxication, and detoxification mechanism in crustaceans under ammonia stress are also discussed.

## 2. Acute Toxicity of Ammonia Nitrogen to Aquatic Crustaceans

The degree of ammonia-induced toxicity to crustaceans depends on many factors, including the water pH level, temperature, and exposure time. Ammonia, nitrite, and nitrate are the three most common dissolved inorganic nitrogen compounds in water [22]. The origin of nitrogenous compounds is principally the decomposition of organic matter. The aerobic nitrification process converts ammonia to nitrite ($NO_2^-$) and then to nitrate ($NO_3^-$) in aquatic environments via *Nitrosomonas* and *Nitrobacter* [23]. Nitrate is considered harmless to fish and can be consumed by plants in water. Generally, $NH_3$ is the most toxic form to crustaceans, followed by $NO_2^-$ and $NO_3^-$ in descending order [18].

### 2.1. Acute Toxic Effects of Ammonia Nitrogen on Shrimp

Numerous studies have been dedicated to discussing the acute toxic effects of ammonia nitrogen on shrimp, as presented in Table 1. Previous studies demonstrated diverse specifications of the same shrimp, which exhibited different tolerances to ammonia. For shrimp in the post-larval (PL) stage, Alcaraz et al. [24] suggested that the acute toxicity of the 48 h $LC_{50}$ of total ammonia nitrogen (TAN) in the white shrimp *Penaeus setiferus* in the PL20 stage was 9.38 mg/L. In the study by Kir and Kumlu [25], the 24, 48, 72, and 96 h $LC_{50}$ of TAN at 15 g/L salinity in the green tiger shrimp *Penaeus semisulcatus* were 14.81 mg/L, 11.18 mg/L, 7.92 mg/L, and 7.09 mg/L, respectively. The $LC_{50}$ of TAN increased as salinity decreased, showing a positive exponential relationship with salinity, indicating that higher salinity makes for a higher survival rate at 15, 20, 25, 30, and 40 g/L salinity levels. As one of the most widely cultivated crustacean species in aquaculture, the Pacific white shrimp *Litopenaeus vannamei* is the highest-yielding crustacean species in the world and can tolerate a wide range of salinities. For the Pacific white shrimp in the post-larval stage, it was demonstrated that the 48 h $LC_{50}$ of TAN in the PL28 shrimp was 39.72 mg/L [26], and the 48 h $LC_{50}$ of TAN in the PL1 shrimp was 13.2 mg/L [27]. Valencia-Castañeda et al. [28] studied acute ammonia nitrogen exposure in the Pacific white shrimp at different salinities.

The LC$_{50}$ of TAN increased when increasing the salinity from 1 g/L to 3 g/L, which proved that higher salinity could increase the ammonia tolerance capacity of the shrimp.

**Table 1.** Summary of acute toxicity of ammonia-N in shrimps.

| Stage | Test Object | Size | Temperature | pH | Salinity | Test Time | LC$_{50}$ Ammonia-N (mg/L) | LC$_{50}$ NH$_3$-N (mg/L) | References |
|---|---|---|---|---|---|---|---|---|---|
| Post-larvae | White shrimp (*Penaeus setiferus*) | PL20 | 28 ± 1 °C | 8.42 ± 0.01 | 25 ± 1 ppt | 24 h<br>48 h<br>72 h | 11.55<br>9.38<br>8.69 | 1.49<br>1.21<br>1.12 | [24] |
| | Green tiger shrimp (*Penaeus semisulcatus*) | 27.5 ± 1.4 mg | 25 °C | 8.2 | 15 g/L | 24 h<br>48 h<br>72 h<br>96 h | 14.81<br>11.18<br>7.92<br>7.09 | 0.73<br>0.55<br>0.39<br>0.34 | [25] |
| | Green tiger shrimp | 27.5 ± 1.4 mg | 25 °C | 8.2 | 25 g/L | 24 h<br>48 h<br>72 h<br>96 h | 31.95<br>16.98<br>12.53<br>8.94 | 1.41<br>0.75<br>0.55<br>0.39 | [25] |
| | Green tiger shrimp | 27.5 ± 1.4 mg | 25 °C | 8.2 | 35 g/L | 24 h<br>48 h<br>72 h<br>96 h | 58.88<br>23.33<br>20.50<br>18.72 | 2.38<br>0.94<br>0.83<br>0.75 | [25] |
| | Pacific white shrimp (*Litopenaeus vannamei*) | PL28 | 28 °C | 7.8 | 10 ppt | 48 h | 39.72 | 2.09 | [26] |
| | Pacific white shrimp | PL42 | 28 °C | 7.8 | 18 ppt | 48 h | 42.9 | 2.26 | [26] |
| | Pacific white shrimp | PL1 | 26 ± 1 °C | 8.5 | 34 g/L | 24 h | 13.2 | 1.94 | [27] |
| | Pacific white shrimp | PL20 | 26.1 ± 0.2 °C | 8.2 ± 0.2 | 1 g/L | 24 h<br>48 h<br>72 h<br>96 h | 38.9<br>14.0<br>11.8<br>10.8 | 3.6<br>1.3<br>1.1<br>1.0 | [28] |
| | Pacific white shrimp | PL20 | 26.1 ± 0.2 °C | 8.2 ± 0.2 | 3 g/L | 24 h<br>48 h<br>72 h<br>96 h | 48.4<br>22.5<br>18.1<br>16.3 | 3.7<br>1.7<br>1.4<br>1.2 | [28] |
| Juvenile | Whiteleg shrimp (*Penaeus vannamei*) | 0.99 ± 0.01 g | 26 °C | 8.08 | 34 g/L | 24 h<br>48 h<br>72 h<br>96 h | 113.4<br>92.5<br>71.2<br>65.2 | NR | [29] |
| | Whiteleg shrimp | 3.8 ± 0.38 g | 23 °C | 7.70 | 34 g/L | 48 h<br>72 h<br>96 h | 110.6<br>85.3<br>70.9 | NR | [29] |
| | Pacific white shrimp | 22 ± 2.4 mm | 23 °C | 8.05 | 15 ‰ | 24 h<br>48 h<br>72 h<br>96 h | 59.72<br>40.58<br>32.15<br>24.39 | 2.95<br>2.00<br>1.59<br>1.20 | [30] |
| | Pacific white shrimp | 22 ± 2.4 mm | 23 °C | 8.05 | 25 ‰ | 24 h<br>48 h<br>72 h<br>96 h | 66.38<br>48.83<br>43.17<br>35.4 | 2.93<br>2.16<br>1.91<br>1.57 | [30] |
| | Pacific white shrimp | 22 ± 2.4 mm | 23 °C | 8.05 | 35 ‰ | 24 h<br>48 h<br>72 h<br>96 h | 68.75<br>53.84<br>44.93<br>39.54 | 2.78<br>2.18<br>1.82<br>1.60 | [30] |
| | White shrimp (*Litopenaeus schmitti*) | 1.5 ± 0.4 g | 20 ± 1 °C | 8.00–8.05 | 5‰ | 24 h<br>48 h<br>72 h<br>96 h | 40.72<br>32.63<br>24.63<br>19.12 | 1.11<br>0.89<br>0.67<br>0.52 | [31] |
| | White shrimp | 1.5 ± 0.4 g | 20 ± 1 °C | 8.00–8.05 | 20‰ | 24 h<br>48 h<br>72 h<br>96 h | 53.52<br>38.60<br>27.76<br>25.55 | 1.28<br>0.92<br>0.66<br>0.62 | [31] |
| | White shrimp | 1.5 ± 0.4 g | 20 ± 1 °C | 8.00–8.05 | 35‰ | 24 h<br>48 h<br>72 h<br>96 h | 54.32<br>47.87<br>41.67<br>38.88 | 1.14<br>1.01<br>0.88<br>0.85 | [31] |

**Table 1.** *Cont.*

| Stage | Test Object | Size | Temperature | pH | Salinity | Test Time | LC$_{50}$ Ammonia-N (mg/L) | LC$_{50}$ NH$_3$-N (mg/L) | References |
|---|---|---|---|---|---|---|---|---|---|
| | Pacific white shrimp | 3.3 g | 26.0 ± 0.5 °C | 8.2 ± 0.2 | 3 g/L | 24 h<br>48 h<br>72 h<br>96 h | 116.3<br>44.6<br>39.6<br>29.0 | 10.1<br>3.9<br>3.4<br>2.5 | [32] |
| | Green tiger shrimp | 1.6 ± 0.2 g | 14 °C | 8.2 | 39 ppt | 24 h<br>48 h<br>72 h<br>96 h | 92.22<br>76.56<br>64.07<br>55.84 | 3.18<br>2.64<br>2.22<br>1.92 | [33] |
| | Green tiger shrimp | 1.6 ± 0.2 g | 18 °C | 8.2 | 39 ppt | 24 h<br>48 h<br>72 h<br>96 h | 57.00<br>46.07<br>41.85<br>36.01 | 2.94<br>2.37<br>2.15<br>1.85 | [33] |
| | Green tiger shrimp | 1.6 ± 0.2 g | 22 °C | 8.2 | 39 ppt | 24 h<br>48 h<br>72 h<br>96 h | 41.10<br>33.42<br>30.61<br>26.72 | 2.77<br>2.25<br>2.06<br>1.80 | [33] |
| | Green tiger shrimp | 1.6 ± 0.2 g | 26 °C | 8.2 | 39 ppt | 24 h<br>48 h<br>72 h<br>96 h | 25.89<br>15.44<br>13.44<br>11.44 | 2.26<br>1.35<br>1.17<br>1.00 | [33] |
| Adult | Signal crayfish (*Pacifastacus leniusculus*) | 35.06 ± 2.06 g | 15 °C | 8.2 | NR | 24 h<br>48 h | 15.0<br>4.9 | 0.91<br>0.30 | [34] |
| | Black tiger shrimp (*Penaeus monodon*) | 15–18 g | 29 ± 0.5 °C | 7.5 ± 0.5 | 29‰ | 96 h | 65 | NR | [35] |
| | Pacific white shrimp | 20.78g | 27.0 ± 0.5 °C | 8.1 ± 0.2 | 30.2‰ | 72 h | 345.94 | NR | [36] |
| | Red swamp crayfish (*Procambarus clarkii*) | 8.47 ± 1.68 g | 24 °C | 7.5 ± 0.3 | NR | 96 h | 135.10 | NR | [37] |
| | Red swamp crayfish | 18.33 ± 2.41 g | 24 °C | 7.5 ± 0.3 | NR | 96 h | 299.61 | NR | [37] |

Note: NR: not reported.

Younger organisms are usually more sensitive to toxins than older ones. For shrimp in the juvenile stage, Frías-Espericueta et al. [29] demonstrated that the 96 h LC$_{50}$ of TAN in juvenile white-leg shrimp was 65.2 mg/L and 70.9 mg/L for small-sized (0.99 ± 0.01 g) and large-sized (3.8 ± 0.38 g) shrimp, respectively. The difference in ammonia toxicity at different salinity levels should be noticed in aquaculture. In the study presented by Lin and Chen [30], the 96 h LC$_{50}$ of TAN in juvenile Pacific white shrimp was 24.39 mg/L, 35.4 mg/L, and 39.54 mg/L when the salinity was 15 ‰, 25‰, and 35‰, respectively, which indicated that the toxicity of ammonia decreased as salinity increased due to the lower uptake of ammonia at high salinities. Similarly, Barbieri [31] also suggested that the LC$_{50}$ of ammonia in the white shrimp *Litopenaeus schmitti* was elevated with increasing salinity, indicating that the tolerance to ammonia stress was enhanced. Valencia-Castañeda et al. [32] investigated the acute toxicity of TAN in the juvenile Pacific white shrimp and found that the 48 and 96 h LC$_{50}$ of TAN were 44.6 mg/L and 29.0 mg/L, respectively. The ammonia toxicity generally increased with the pH level and temperature in the solution due to the increased NH$_3$-N proportion. Kir et al. [33] demonstrated that the 96 h LC$_{50}$ of TAN in juveniles of the green tiger shrimp was 55.84 mg/L, 36.01 mg/L, and 26.72 mg/L at 14 °C, 18 °C, and 22 °C, respectively. The data clearly indicated that the tolerance of green tiger shrimp juveniles to ammonia increased with a decrease in water temperature.

For shrimp in the adult stage, Harris et al. [34] studied the acute toxicity of ammonia in the signal crayfish *Pacifastacus leniusculus*. The 24 and 48 h LC$_{50}$ of TAN were 15.0 mg/L and 4.9 mg/L, respectively. Ammonia nitrogen is commonly considered the primary factor that can rapidly cause increased shrimp mortality. It was reported that the 96 h LC$_{50}$ of

TAN in the adult black tiger shrimp *Penaeus monodon* was 65 mg/L [35], and the 72 h $LC_{50}$ of TAN in the adult Pacific shrimp was 345.94 mg/L [36]. Furthermore, our previous study demonstrated that the red swamp crayfish *Procambarus clarkii* with a larger size exhibited a higher and stronger antioxidant capacity, thus exerting a higher tolerance to ammonia stress [37]. The above studies demonstrated that ammonia-induced toxicity in shrimp varies with pH, temperature, salinity, and the development stage of the organisms.

### 2.2. Acute Toxic Effects of Ammonia Nitrogen on Crabs

Studies dedicated to the acute toxic effects of ammonia nitrogen on crabs in different development stages are summarized in Table 2. As larval stages support population recruitment, it is ecotoxicologically relevant to know what effects ammonia has on crab larvae. Neil et al. [38] studied the acute toxicity of ammonia in mud crab (*Scylla serrata*) larvae. The 24 and 48 h $LC_{50}$ of TAN were 51.51 mg/L and 37.52 mg/L, respectively. Liao et al. [39] suggested that the 24 and 48 h $LC_{50}$ of TAN in blue swimming crab (*Portunus pelagicus*) larvae were 51.04 mg/L and 38.72 mg/L, respectively. However, Diodato et al. [40] found that the 96 h $LC_{50}$ of TAN was 107.97 mg/L in larvae of the southern king crab *Lithodes santolla*, a cold and cold–temperate water species, which indicated that ammonia-induced toxicity might be associated with water temperature.

**Table 2.** Summary of acute toxicity of ammonia-N in crabs.

| Stage | Test Object | Size | Temperature | pH | Salinity | Test Time | $LC_{50}$ Ammonia-N (mg L$^{-1}$) | $LC_{50}$ NH$_3$-N (mg L$^{-1}$) | References |
|---|---|---|---|---|---|---|---|---|---|
| Larvae | Mud crab (*Scylla serrata*) | NR | 28.3 °C | 8.01 | 28 g/L | 24 h<br>48 h | 51.51<br>37.52 | NR | [38] |
| | Blue swimming crab (*Portunus pelagicus*) | NR | 28 ± 0.5 °C | Artificial seawater | Artificial seawater | 24 h<br>48 h<br>72 h<br>96 h | 51.04<br>38.72<br>16.90<br>11.16 | NR | [39] |
| | Southern king crab (*Lithodes santolla*) | NR | 7.5 ± 0.5 °C | 8.15 | 23–24‰ | 96 h | 107.97 | 1.93 | [40] |
| Juveniles | Chinese mitten crab (*Eriocheir sinensis*) | 0.059 ± 0.007 g | 22 ± 0.5 °C | 7.81 ± 0.07 | 30‰ | 24 h<br>48 h<br>72 h<br>96 h | 109.3<br>60.9<br>45.3<br>31.6 | 3.10<br>1.73<br>1.29<br>0.9 | [41] |
| | Chinese mitten crab | 3.16 ± 0.91 g | 20 ± 1 °C | 7.45 ± 0.08 | 30‰ | 24 h<br>48 h<br>72 h<br>96 h | 251.58<br>217.61<br>156.05<br>119.67 | 2.75<br>2.37<br>1.70<br>1.31 | [42] |
| | Chinese mitten crab | 8.19 ± 0.78 g | 22.8 °C | 7.74 | Freshwater | 96 h | 90.92 | 2.32 | [43] |
| | Blue swimmer crab | 0.002 ± 0.001 g | 28 °C | 8.1 | 30‰ | 12 h<br>24 h<br>36 h<br>48 h<br>60 h<br>72 h<br>84 h<br>96 h | NR | 5.50<br>3.16<br>2.40<br>1.93<br>1.85<br>1.79<br>1.78<br>1.65 | [44] |
| | Blue swimmer crab | 0.028 ± 0.001 g | 28 °C | 8.1 | 30‰ | 12 h<br>24 h<br>36 h<br>48 h<br>60 h<br>72 h<br>84 h<br>96 h | NR | 6.48<br>3.59<br>2.78<br>2.31<br>2.07<br>1.93<br>1.87<br>1.80 | [44] |
| | Blue swimmer crab | 0.187 ± 0.002 g | 28 °C | 8.1 | 30‰ | 24 h<br>36 h<br>48 h<br>60 h<br>72 h<br>84 h<br>96 h | NR | 6.19<br>4.97<br>3.59<br>3.33<br>2.90<br>2.83<br>2.68 | [44] |

**Table 2.** *Cont.*

| Stage | Test Object | Size | Temperature | pH | Salinity | Test Time | LC$_{50}$ Ammonia-N (mg L$^{-1}$) | LC$_{50}$ NH$_3$-N (mg L$^{-1}$) | References |
|---|---|---|---|---|---|---|---|---|---|
| | Blue swimmer crab | 0.732 ± 0.036 g | 28 °C | 8.1 | 30‰ | 24 h 36 h 48 h 60 h 72 h 84 h 96 h | NR | 7.70 6.34 4.89 4.44 4.00 3.91 3.62 | [44] |
| | Mud crabs | 0.373 ± 0.024 g | 28 °C | 8.10 | 30‰ | 24 h 36 h 48 h 60 h 72 h 84 h 96 h | NR | 9.65 8.86 7.86 7.43 7.15 7.00 6.81 | [45] |
| Adults | Blue crab (*Scylla paramamosain*) | 29.41 ± 4.12 g | 27.17 ± 0.14 °C | 8.16 ± 0.06 | 14.62 ± 0.35‰ | 24 h 48 h | 104.793 66.124 | 8.396 5.298 | [46] |
| | Swimming crab (*Portunus trituberculatus*) | 206.8 ± 7.5 g | 15.2 ± 0.6 °C | 7.8 ± 0.3 | 30.8 ± 0.5‰ | 96 h | 40 | NR | [47] |

Note: NR: not reported.

For crabs in the juvenile stage, Zhao et al. [41] suggested that the 48 and 96 h LC$_{50}$ of TAN in Chinese mitten crab (0.059 ± 0.007 g) at 30‰ salinity were 60.9 mg/L and 31.6 mg/L, respectively. Hong et al. [42] also studied the acute toxicity of ammonia nitrogen in Chinese mitten crab (3.16 ± 0.91 g) at 30‰ salinity. The 48 and 96 h LC$_{50}$ of TAN were 217.61 mg/L and 119.67 mg/L, which implies that crabs with larger sizes may exert a stronger tolerance to ammonia. Wang et al. [43] found that the 96 h LC$_{50}$ of TAN in Chinese mitten crab (8.19 ± 0.78 g) in freshwater was 90.92 mg/L, which indicated that higher salinity could enhance the ammonia nitrogen tolerance of crabs. Romano and Zeng [44] investigated the acute toxicity of ammonia nitrogen in juvenile blue swimming crabs with different sizes. The results suggested that the 24, 48, and 96 h LC$_{50}$ of TAN increased with the crab size, showing that crabs with a larger size showed a stronger ammonia nitrogen tolerance. Romano and Zeng [45] also studied the acute toxicity of ammonia nitrogen in juvenile mud crabs. The 48 and 96 h LC50 of TAN were 7.86 mg/L and 6.81 mg/L, respectively.

As for crabs in the adult stage, Peng et al. [46] found that the 24 and 48 h LC$_{50}$ of TAN in the blue crab *Scylla paramamosain* were 104.793 mg/L and 66.124 mg/L, respectively. Meng et al. [47] studied the acute toxicity of ammonia nitrogen in the adult swimming crab *Portunus trituberculatus* (206.8 ± 7.5 g). The 96 h LC$_{50}$ of TAN was 40 mg/L, which proved that larger-sized crabs possessed a stronger ammonia nitrogen tolerance capacity.

## 3. Potential Mechanisms of Ammonia-Induced Acute Toxicity

### 3.1. Cellular Uptake of Ammonia Nitrogen

In order to exert acute toxicity, sufficient concentrations of ammonia must enter aquatic organisms. In fish, ammonia can enter the body through its gill epithelium, almost exclusively via passive diffusion [48]. Previous studies have also demonstrated that a high NH$_3$ level is toxic and deadly for crustaceans, and the lethality of NH$_3$ is attributed to its quick transfer across gill epithelia and its accumulation therein [49,50]. In crustaceans, ammonia mainly penetrates the gill epithelium as NH$_3$, which can result in ammonia accumulation in the hemolymph [51]. Tsui et al. [52] proposed that high external NH$_4^+$ could inhibit the Na$^+$/NH$_4^+$ exchange process, causing osmoregulatory failure and sodium depletion. Studies have also demonstrated that acute ammonia exposure can result in the inhibition of Na$^+$ influx in the goldfish *Carassius auratus* [53] and the rainbow trout

*Oncorhynchus mykiss* [54]. Therefore, ammonia can result in the disruption of gill epithelial integrity and cause adverse effects on ion transport and other cellular processes.

### 3.2. Accumulation and Transformation of Ammonia

The majority of fishes are ammoniotelic, producing and excreting ammonia by diffusing $NH_3$ through the gills [55]. Similarly, most crustaceans are also ammoniotelic, and their nitrogenous metabolic products are excreted mainly in the ammonia form, with a minor part in the form of urea [56]. Exposure to high ambient ammonia concentrations or the inhibition of endogenous excretion could result in ammonia accumulation in aquatic animals [57]. Ammonia accumulated in the fish body is toxic, and the consequence of exposure affects various cellular and organismal processes [58]. Compared to fish, crustaceans exhibit a higher tolerance to ammonia, which can be attributed to the accommodation of high concentrations of blood ammonia and therefore a reduction in the net $NH_3$ influx [59].

The most susceptible organ of crustaceans is the gills, and they play a vital role in various physiological processes, including osmoregulation, acid/base balance, and ammonia-N excretion [60,61]. Ammonia-induced toxicity in crustaceans under elevated nitrogenous waste exposure is largely dependent on their hemolymph ammonia accumulation. In order to lower the ammonia accumulation internally, aquatic animals can reduce ammonia excretion by decreasing endogenous ammonia production from amino acid catabolism [18]. A significant increase in the total free amino acid contents was also observed in the swamp eel *Monopterus albus* under ammonia stress, which indicated a decrease in the amino acid catabolism rate [62]. Increased urea and reduced amino acid catabolism were noticed in the hepatopancreas of the swimming crab, indicating that the urea cycle was utilized to transform ammonia, and amino acid catabolism was inhibited to reduce endogenous ammonia generation [47]. Previous studies also reported that the hepatopancreas can exert a detoxification effect by transforming ammonia into urea in the white shrimp [63] and mud crab [20] under ammonia stress. The above studies indicated that the accumulation and transformation of ammonia determine the intoxication as well as detoxification of ammonia-induced toxicity in aquatic crustaceans.

### 3.3. Oxidative Stress

Ammonia accumulation can result in detrimental physiological and biochemical alterations in aquatic animals, which are mainly characterized by a decreased blood oxygen-carrying capacity [64]. Anoxic conditions will further increase ammonia concentrations via the entrance of excessive ammonia into the body, subsequently resulting in reactive oxygen species (ROS) production and oxidative stress, which is highly toxic [20,65]. To counter the deleterious effects caused by excessive ROS, aquatic animals have developed an antioxidant defense system comprising catalase (CAT), superoxide dismutase (SOD), and glutathione peroxidase (GPx) [66]. The fluctuation of antioxidant indexes in the hemolymph and hepatopancreas can effectively reflect the health of aquatic crustaceans [67].

Several studies demonstrated that ammonia accumulation in the tissues of aquatic animals can trigger ROS release [68,69]. In crustaceans, the antioxidant system plays a vital role in the oxidative defense system by scavenging ROS [70,71]. SOD is considered an important indicator of the cell oxidant/antioxidant balance in protecting against cellular damage [42]. It was reported that the hepatopancreatic SOD activity in the white shrimp was significantly diminished after exposure to ammonia [14]. The hemolymph SOD and GPx activities also exhibited a significant decrease in the Chinese mitten crab under ammonia stress [72]. Similarly, our previous study demonstrated that SOD and CAT in the hepatopancreas exhibited a significant decrease under ammonia stress, which implies that the antioxidant capacity of the red swamp crayfish was inhibited [37].

Decreased antioxidant activities indicate a loss in a compensatory mechanism due to excessive ROS. ROS might damage the cell membranes by forming lipid peroxides [73]. The lipid peroxidation level, measured as malonaldehyde (MDA), showed a significant increase in the white shrimp after ammonia exposure, indicating that ammonia exposure

disrupts the oxidant and antioxidant balance [74]. Ammonia exposure could result in oxidative stress by inhibiting antioxidant enzyme activities and elevating MDA production, eventually resulting in the high mortality of aquatic crustaceans.

### 3.4. Innate Immune Dysfunction

Innate immunity, comprising both humoral components and cellular activities, is considered the primary defense mechanism in crustaceans and aims to eliminate potentially hazardous foreign organisms [75,76]. Ammonia can disrupt crustaceans' immunity, resulting in physiological disturbances, including molting failure, slower growth, and increased mortality [24]. One of the most important immunological responses for crustaceans is the production and release of hemocytes into the hemolymph [77]. Hemocytes, including granular cells, hyaline cells, and semi-granular cells, are the major cellular components participating in phagocytosis and antimicrobial peptide production in crustaceans [78,79].

Crustacean penaeidins are major antimicrobial proteins in their humoral innate defense [80]. C-type lectin can play a crucial part in pathogen recognition and innate immunity in shrimp [81]. Lysozymes are also considered a strong barrier to bacterial pathogen invasion in crustaceans' innate immunity [82]. Studies have suggested that excessive ammonia inhibits the immunity of the freshwater prawn *Macrobrachium rosenbergii*, therefore increasing the risk of infection from microbial pathogens [83]. Similarly, Yeh et al. [84] demonstrated that phagocytic activity was decreased in the white shrimp following ammonia exposure. In addition, ammonia also weakens the immunity of the white shrimp, including decreased phagocytic activity and antibacterial activity, making it more susceptible to white spot syndrome virus [85]. Li et al. [19] demonstrated that multiple immune-function-related genes, including G protein-coupled receptor, toll-like receptor, and C-type lectin, were significantly down-regulated by ammonia exposure, which confirmed the immune-impairing effects. Acute ammonia exposure can induce innate immune dysfunction, eventually resulting in high mortality. Alterations in innate immune molecule parameters can be identified as reliable indicators to evaluate the health status of crustaceans.

## 4. Current Research Gaps and Future Directions

### 4.1. Excretion of Ammonia Nitrogen

Ammonia can be excreted by aquatic organisms via the gills and other epithelia as nitrogenous waste. Previous studies demonstrated that crustaceans can resist high ammonia levels through an ammonia excretion strategy to maintain normal cellular functions [86,87]. The gills and the antennal gland are important organs involved in the excretory process in crustaceans, and nearly 60–95% of the total excreted nitrogen is excreted through the gills [13,88]. Kormanik and Cameron [89] suggested that ammonia excretion in the blue crab *Callinectes sapidus* occurred mainly by $NH_3$ diffusion. However, another researcher showed that ammonia excretion in the shore crab *Carcinus maenas* is partially in the $NH_4^+$ form [90]. Hong et al. [42] found an increase in hemolymph urea in the juvenile Chinese mitten crab, suggesting that urea synthesis is an acute response under high ammonia stress.

The ammonia excretory capacity of crustaceans mainly relies on transporters, including $Na^+/K^+$-ATPase, $K^+$-channels, Rhesus-like proteins, V-type $H^+$-ATPase, $Na^+/H^+$ exchangers, ammonia transporters, and aquaporins, except for the passive transport process [13,91]. Crustaceans exhibit various ammonia excretion mechanisms, which make them a good model to investigate nitrogen excretion. The majority of aquatic animals, especially crustaceans, are ammonotelic, but a small number of ureotelic species have also been reported [92]. Whether ammonia is mainly excreted via $NH_3$ diffusion or $NH_4^+$ transport in crustaceans still remains to be elucidated.

### 4.2. Ammonia-Induced Neurotoxicity

Ammonia is a potent neurotoxin that can cause adverse effects on the nervous system in aquatic animals, including hyperexcitability, hyperventilation, loss of equilibrium, coma, and even death [93]. In fish, excessive ammonia accumulation results in the activation of the

N-methyl-D-aspartic acid (NMDA)-type glutamate receptor, which in turn causes damage to the central nervous system, leading to cell death [57]. Tudorache et al. [94] suggested that fast-start performance and predation behaviors were also affected by ammonia exposure in the brown trout *Salmo trutta*.

In crustaceans, previous studies concerning ammonia-induced neurotoxicity mainly focus on abnormal alterations in neurohormones, including crustacean hyperglycemia hormone (CHH) and dopamine (DA). CHH is a type of neural factor released in response to environmental stress, which can help white shrimp better adapt to a high environmental concentration of ammonia nitrogen [92]. DA is also an important neuroendocrine factor that has been identified in crustaceans [95,96]. To our knowledge, ammonia can induce various abnormal physiological functions, including mitochondrial damage, inflammatory response dysfunction, and memory impairment. However, the underlying ammonia-induced neurotoxic effects on crustaceans are still lacking and need to be investigated.

### 4.3. Threat to Freshwater Aquaculture

Previous studies concerning ammonia toxicity were mainly focused on crustaceans in the seawater environment. Freshwater aquaculture ponds are relatively static, in a closed environment, and are greatly affected by outside activities. As global aquaculture is changing from extensive to intensive systems, almost 70% of the nitrogen is released directly into the aquaculture environment [97]. Ammonia is becoming one of the most important indexes in intensive fish farming, which can directly impact fish survival [98] and aquaculture productivity [99]. Various diseases caused by ammonia have blossomed with the development of intensive aquaculture, resulting in economic losses in the aquaculture of crustaceans.

In an aquatic environment, the metabolic waste of aquatic animals and the decomposition of residual feeds are two primary sources of ammonia [100]. Ammonia can be taken up by phytoplankton, assimilated again by filter-feeding animals, and oxidized by ammonia-oxidizing microorganisms [101]. In traditional aquaculture, changing water is an effective way to lower ammonia levels. With the development of fishery facilities and aquaculture techniques, recirculating aquaculture systems and the biofloc culture are becoming reliable farming models for controlling excessive ammonia and maintaining water quality [102,103]. Multiple approaches should be utilized to maintain low ammonia levels in the aquaculture environment, which is essential for the health of cultured animals.

### 5. Conclusions

Excessive ammonia levels can adversely affect the physiological conditions of aquatic crustaceans, which has become a serious problem with increasing anthropogenic activities. We present a comprehensive review compiling existing research concerning acute ammonia toxicity in shrimp and crabs. These studies exhibited a tight association between the developmental stages of crustaceans and ammonia-induced toxicity. The toxic mechanisms might be attributed to decreased antioxidant capacity and innate immunity. Hence, investigating the acute toxicity and the underlying mechanisms of ammonia is crucial for the aquaculture of crustaceans.

**Funding:** This work was funded by the National Natural Science Foundation of China (No. 42207334), the Youth Project of Natural Science Foundation of Hunan Province (2022JJ40289), the Excellent Youth Project of Hunan Education Department (21B0613), the Doctoral Start-Up Project of Hunan University of Arts and Science (21BSQD02), the Innovation and Entrepreneurship Training program for College Students (Education Department of Hunan Province [2021] No. 3225), and the Aquaculture Germplasm Resources Investigation Project in Hunan Province (Hunan Provincial Department of Finance [2021] NO.37). The APC was funded by the Aquaculture Germplasm Resources Investigation Project in Hunan Province (Hunan Provincial Department of Finance [2021] NO.37).

**Acknowledgments:** We would like to thank anonymous reviewers for their very helpful and constructive comments that greatly improved the manuscript. We also thank Ana Gavrilović from Faculty of Agriculture, University of Zagreb for her helpful comments.

**Conflicts of Interest:** The authors declare no competing financial interests.

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
