# Peer review of "A Review of the Emerging Risks of Acute Ammonia Nitrogen Toxicity to Aquatic Decapod Crustaceans"

_water, doi:10.3390/w15010027_

Round 1
Reviewer 1 Report
The authors examined the mechanisms concerning ammonia-induced acute toxicity and the relationship between ammonia stress and the defense responses in shrimp and crabs. This review mentioned that the degree of ammonia-induced toxicity in crustaceans depends on many factors, summarized in Table 1.
Main observations:
To quickly visualize these comparative data (Table 1), one should also have a graphic comparing these factors with the stage (post-larvae, juvenile, and adult shrimps). It is interesting because the authors reported that adult crayfish with a larger size exhibited a higher and stronger antioxidant capacity (lines 203-205).
Reviewer 2 Report
Review of manuscript: A review of the emerging risks of acute ammonia nitrogen toxicity on aquatic decapod crustaceans, submitter to Water journal
This review article represents an interesting document considering acute toxicity of ammonia to aquaculture in general and crustaceans in particular in a situation of lacking data and approaches. The issue emerges from the fact that ammonia accumulates easily in aquatic systems because it is a natural byproduct of fish metabolism. Because it is continuously excreted, producers and other fish owners must regularly measure ammonia and eliminate it from systems before it can accumulate and damage or kill fish. Additionally, ammonia may be present in source waters such as some municipal (city) or well waters.
I recommend the followings:
# The Abstracts (line 28) starts with sentence “Waterborne ammonia is becoming one of the most notorious pollutants in aquatic habitats, and has been shown to induce a range of eco-toxicological effects on aquatic animals. This is fine, but along with the role of aquaculture intensive system, please insert here the role of wastewater treatment and agriculture systems.
# The graphical abstract does not reflects any novelty, it is a simple pathway.
# At the line 14, the Figure 1. Conversion of aquatic crustaceans-generated ammonia in the 140 water (modified from Francis-Floyd, 2009), is no needed.
# References within texts and list need to be revised, eg. On page 5 it stands Francis-Floyd, 2009, while in the list stands: Francis-Floyd, R., Watson, C., Petty, D., Pouder, D. B. 2009. Ammonia in aquatic systems. EDIS.
Pls. follow the journals guideline.
